# Molecular Characterization of *Entamoeba* spp. in Pigs with Diarrhea in Southern China

**DOI:** 10.3390/ani12141764

**Published:** 2022-07-09

**Authors:** Pei Wang, Sen Li, Yang Zou, Ru-Yi Han, Ping Wang, De-Ping Song, Cheng-Bin Wang, Xiao-Qing Chen

**Affiliations:** 1College of Animal Science and Technology, Jiangxi Agricultural University, Nanchang 330045, China; 15779520069@163.com (P.W.); lisentdcq@163.com (S.L.); hhhry5201314@163.com (R.-Y.H.); jxjs6263wplm@163.com (P.W.); sdp8701@jxau.edu.cn (D.-P.S.); chengbinwang@jxau.edu.cn (C.-B.W.); 2State Key Laboratory of Veterinary Etiological Biology, Key Laboratory of Veterinary Parasitology of Gansu Province, Lanzhou Veterinary Research Institute, Chinese Academy of Agricultural Sciences, Lanzhou 730046, China; zouyangdr@163.com

**Keywords:** *Entamoeba* spp., zoonotic, diarrheic pigs, southern China

## Abstract

**Simple Summary:**

China is an important country in the world for pig breeding and pork consumption, yet only two provinces have reported *Entamoeba* spp. infections. In this study, fecal samples of pigs with diarrhea were collected from three provinces in southern China, and *Entamoeba* spp. infection was detected by nested PCR. The results showed that the total infection rate was 58.4%. *Entamoeba polecki* and *Entamoeba suis* were detected. The ST1 and ST3 subtypes of *Entamoeba polecki* were detected, and a relatively serious mixed infection was found, with the most common form of mixed infection being *Entamoeba polecki* ST1 + *Entamoeba polecki* ST3. These findings provide baseline data for preventing and controlling *Entamoeba* spp. infection in southwestern China.

**Abstract:**

*Entamoeba* spp. is a common zoonotic intestinal protozoan that can parasitize most vertebrates, including humans and pigs, causing severe intestinal diseases and posing a serious threat to public health. However, the available data on *Entamoeba* spp. infection in pigs are relatively limited in China. To characterize the infection of *Entamoeba* spp. within pigs in southern China, 1254 fecal samples of diarrheic pigs were collected from 37 intensive pig farms in Hunan, Jiangxi and Fujian provinces and the infection of *Entamoeba* spp. was investigated based on the small subunit rRNA (SSU rRNA) gene. The overall infection rate of *Entamoeba* spp. was 58.4% (732/1254), including 38.4% (118/307) in suckling piglets, 51.2% (153/299) in weaned piglets, 57.9% (55/95) in fattening pigs and 73.4% (406/553) in sows, respectively. Moreover, age and the sampling cities in Jiangxi and Fujian provinces were found to be the key factors influencing the infection of *Entamoeba* spp. (*p* < 0.05). Two subtypes (ST1 and ST3) with a zoonotic potential of *Entamoeba polecki* and *Entamoeba suis* were detected in all age groups of pigs and all sampling areas, with the predominant species and predominant subtype being *E. polecki* (91.3%, 668/732) and *E. polecki* ST1 (573/668), respectively, and *E. polecki* ST1 + *E. polecki* ST3 (78.6%, 239/304) being the most frequently detected form of mixed infection. Severe *Entamoeba* spp. infection and zoonotic subtypes were found in this study, exposing a large public health problem in the study area, and strategies need to be implemented to eliminate the risk in the future.

## 1. Introduction

*Entamoeba* spp. can be divided into parasitic species that infect humans and animals, and free-living species that live in water and mud with the occasional invasion of animals, depending on the living environment [1,2]. Except for *Entamoeba gingivalis*, which does not form cysts, most *Entamoeba* spp. species have cyst or trophozoite stages during their whole life cycle [3,4]. Like other intestinal protozoa, the fecal–oral route is an important route of transmission for *Entamoeba* spp. [5]. Up to now, seven (*E. bangladeshi*, *E. histolytica*, *E. dispar*, *E. coli*, *E. moshkovskii*, *E. hartmanni*, and *E. polecki*) and three (*E. polecki*, *E. suis* and *E. histolytica*) species of *Entamoeba* spp. have been identified in humans and pigs, respectively [6,7,8]. Most of them are considered harmless, but some species can cause serious intestinal diseases [9,10]. Sometimes, *E. histolytica* causes more serious diseases, and relatively more research has been conducted concerning *E. histolytica* worldwide, and hygiene conditions are closely related to the disease [6,11]. *E. histolytica* can invade not only the liver and colon tissue of the host, causing colon ulcers and liver abscesses, but also the lungs, skin, genitourinary tract, brain and spleen, causing other severe diseases [6,12]. This may explain why amebiasis is one of the three deadliest diseases in the world, infecting approximately 50 million people worldwide and causing more than 50,000 deaths each year, for which countries have to pay high public health care costs every year [12,13,14,15]. Even worse, *E. histolytica* has great transmissibility, requiring only one normally infective oocyst to cause amoebiasis in the host [3,13]. Although *E. histolytica* has not been observed in naturally raised pigs to date, some studies have found that pigs can be infected with *E. histolytica* with human intervention, and thus pigs are considered to be a potential reservoir of pathogens [3,7,14,16,17]. The high population density in China and the presence of intensive domestic pig farms undoubtedly increases the risk of human contact with pigs and thus infection with pathogens. Humans are likely to be infected with *Entamoeba* spp. through pigs. However, although *Entamoeba* spp. infections have been reported in several studies in China, the main focus has been on investigating infections in humans and non-human primates [18]. Some researchers have found that although monkeys were at greater risk of an *Entamoeba* spp. infection, the species of *Entamoeba* parasitized in different species of monkeys were different [19]. Patients with AIDS and homosexuals were found to be at higher risk of infection [20]. Therefore, investigation of *Entamoeba* spp. infection in pigs is necessary for improving public health safety and studying the genetic structure.

*E. suis* and *E. polecki* were the two *Entamoeba* spp. species most frequently detected in pigs [16,17,21,22,23,24]. Although *E. suis* has previously been considered host-specific with parasitism only in pigs, later studies detected *E. suis* in gorillas, so further studies are needed to determine whether *E. suis* is a strictly host-specific *Entamoeba* spp. species [2,25]. *E. suis* invades the lamina propria of the host, with hemorrhagic colitis being the typical pathological feature [3,22]. *E. polecki* has been detected in a variety of hosts, including humans and pigs, with zoonotic potential [2,14,25]. Moreover, there is also a degree of genetic variation in *E. polecki*, which can be classified into four subtypes (ST1–ST4) based on the small subunit ribosomal RNA (SSU rRNA) genes [14,25]. These four subtypes have all been observed in humans, while only ST1 and ST3 have been observed in pigs and ST2 has only been detected in non-human primates [2,14,25,26]. Not until 2016, when ST2 was detected in wild Celebes crested macaques, it was realized that ST2 was not parasitic only in humans [25,27]. Though swine single infection with *E. polecki* was not pathogenic, mixed infection with certain pathogenic bacteria will aggravate the disease [8,9,28,29,30].

*Entamoeba* spp. infection in pigs has been published in thirteen countries, but the microscopic examination was the method used in most reports [14,22,25,28,29,30]. However, significant genetic diversity may exist in some morphologically undifferentiated *Entamoeba* species and some morphological differences do not reflect species-level differences [3,8,31]. Based on the SSU rRNA gene of *Entamoeba* spp., several pairs of relatively mature primers are now available for the species/subtypes identification of *Entamoeba* spp. [3,4,7,14,26,32]. Although these primers have only been developed in recent years, molecular characterization reports of *Entamoeba* spp. infections in pigs were very limited. For China, a critical pig breeding country, there were only two reports of *Endamoeba* spp. infections in pigs based on molecular means [14,25]. This study aims to investigate the infection of *Entamoeba* spp. in diarrheic pigs in southern China that will supplement the data on *Entamoeba* spp. infection in pigs, which will deepen the study of the genetic diversity of *Entamoeba* spp. and may develop more effective preventive control measures.

## 2. Materials and Methods

### 2.1. Ethics Statement

This experiment was conducted in strict accordance with the experimental animal regulations of Jiangxi Agricultural University. Sampling was carried out only after obtaining the consent of the farm owner, and the whole sampling process did not cause damage to animals.

### 2.2. Specimen Collection

Fresh fecal specimens were collected from 1254 pigs with diarrhea in the Jiangxi, Hunan and Fujian provinces during 2015–2019, with 553 from sows, 307 from suckling pigs, 229 from weaned pigs and 95 from fattening pigs (Figure 1 and Table 1). These 1254 samples were collected from 37 large intensive pig farms under scientific control, 23 of which were located in Jiangxi provinces, 6 in Hunan provinces, and the remaining 8 in Fujian provinces. The number of samples collected from farm to farm was variable, ranging from 1 to 141. Only pigs excreting thin feces or showing watery diarrhea were sampled, with anal swabs being the only sampling method used in this study. Each sample was clearly labeled with information on age, collection location and sampling time before being rapidly frozen and finally stored at −80°C until DNA extraction.

### 2.3. DNA Extraction

To remove the impurities, each specimen (approximately 300 mg) was washed three times with distilled water by centrifugation at 13,000 g for five minutes. According to the manufacturer’s instructions, genomic DNA was extracted from the remaining sediments by the commercial E.Z.N.A.^®^ Fecal DNA Kit (D4015-02, Omega Bio-Tek Inc., Norcross, GA, USA) and stored at −20 °C until used for molecular analysis.

### 2.4. PCR Amplification

To screen the species/subtypes of *Entamoeba* spp., seven primer pairs were used for nested PCR, which was designed based on the small subunit ribosomal RNA (SSU rRNA) genes of *Entamoeba* spp. To detect the infection of *E. histolytica*, the first round of PCR was performed with primers E-1 and E-2, and the second round of PCR was performed with primers EH-1 and EH-2 with a target gene fragment of 439 bp. For *E.*
*suis*, the final amplified target gene size was 320 bp by first-round PCR using primers 764-RD3 and second-round PCR using primers 764–765. Eploec F6 and Eploec R6 were used for the first round of PCR reaction to amplify a 430 bp gene of *E. polecki*. Then the second round of PCR for subtype-specific characterization of *E. polecki* used primers Epolecki 1-Epolecki 2 (ST1, 201 bp) and EpST3F1-EpST3R2 (ST3, 190 bp) using the primary PCR products as a template was carried out, respectively [14,25]. Details of all primers in this study were listed in Table 2.

All PCR reaction systems in this study were 25 µL. For *E. histolytica*, the reaction system contained 0.2 mM dNTP, 1 mM 10 × Taq buffer (Mg^2+^ free), 1.5 mM MgCl_2_, 0.4 mM each primer, 2 µL of genomic DNA and, finally, 25 µL was made up with reagent water. The difference between *E. suis* and *E. histolytica* was 2 mM MgCl_2_, 0.5 mM of each primer and 1 µL of genomic DNA, while *E. polecki* was only 1 µL of genomic DNA compared to *E. histolytica*. The annealing temperatures of all PCR reactions involved in this experiment were listed in Table 2. The initial denaturation time of all PCR reactions (except *E. suis*) was 94 °C for 5 min, the second step was 94 °C for 30 s, followed by 45 s of annealing, then 72 °C extensions for 1 min, which were to be repeated 35 times from the second step to the extension and the end extension for 7 min. For *E. suis*, the annealing time was extended to 1 min. The secondary PCR products were examined by electrophoresis in 1% (*w*/*v*) agarose gels containing ethidium bromide.

### 2.5. Sequence and Phylogenetic Analyses

All PCR products that could amplify the target band size were sent to Hunan Tsingke Biotechnology Co., Ltd. (Changsha High-Tech Development District, Changsha China), for bi-directional sequencing. All sequenced sequences were calibrated with Chromas V.2.6.4 before splicing, and finally, the spliced sequences were aligned with the sequences in NCBI http://www.ncbi.lm.nih.gov/GenBank (accessed on 15 May 2021). Based on the results of alignment, it was determined whether the samples were infected with *Entamoeba* spp. and the species/subtypes of *Entamoeba* spp. Representative sequences of *E. polecki* ST1, *E. polecki* ST3 and *E. suis* obtained in this study have been deposited in the GenBank database under accession numbers MZ203209-MZ203510 (*E. polecki* ST1), MZ203512-MZ203514 (*E. polecki* ST3) and MZ203516-MZ203518 (*E. suis*), respectively. Phylogenetic analysis was performed based on the SSU rRNA gene of *Entamoeba* spp. using the auxiliary software MEG7 http://www.megasoftware.net/ (accessed on 16 March 2022) by the neighbor-joining method (NJ), with mode set to *p*-distance and bootstrap, and the bootstrap parameter set to 1000. AF149905 and AF149908 were set as the out-groups. For the bootstrap parameter, values of less than 50 will not appear on the branch.

### 2.6. Statistical Analysis

In this study, the chi-square (χ^2^) test of SPSS version 25.0 (IBM SPSS Inc., Chicago, IL, USA) was used to analyze the correlation between *Entamoeba* spp. infection and independent factors (region and age), and the difference of infection was considered to be significantly correlated with this factor when the *p*-value was less than 0.05. The infection risk of *Entamoeba* spp. was also assessed in diarrheic pigs from different regions and ages, with odds ratios (ORs), and 95% confidence intervals (Cls) were used to ensure the accuracy of the results.

## 3. Results

### 3.1. Prevalence of Entamoeba spp. in Diarrheic Pigs

The epidemiological investigation of 1254 diarrheic pigs’ fecal samples for *Entamoeba* spp. revealed that 732 samples were positive for *Entamoeba* spp. with an overall infection rate of 58.4% (732/1254, 95%Cl 55.64–64.10). The infection rates in Hunan, Jiangxi and Fujian provinces were 59.0% (67/135, 95%Cl 41.20–58.06), 59.5% (616/1036, 95%Cl 56.47–62.45) and 49.6% (49/83, 95%Cl 48.46–69.62), respectively, whereas the infection rates of *Entamoeba* spp. varied widely among pigs of different ages, with the highest rate being in sows at 73.4% (406/553, 95%Cl 69.74–77.10), followed by fattening, weaned and suckling piglets at 57.9% (55/95, 95%Cl 47.97–63.82), 51.2% (153/299, 95%Cl 45.50–56.84) and 38.4% (118/307, 95%Cl 33.00–43.88), respectively. The differences in infection in different age groups were statistically significant (*p* < 0.001). Notably, the infection risk of *Entamoeba* spp. in sows was 4.42 times (95%Cl 3.29–5.96) higher than in suckling piglets (Table 3).

*Entamoeba* spp. was detected in all sampling cities, with the highest infection observed (75.4%, 178/236, 95%Cl 69.93–80.92) in Yichun City, Jiangxi Province, and the lowest (37.5%, 9/27, 95%Cl 18.13–56.87) found in Zhuzhou City, Hunan Province. The differences in infection between the seven sampling cities in Jiangxi Province and the two sampling cities in Fujian Province were statistically significant (*p* < 0.05). The infection risk of *Entamoeba* spp. in diarrheic pigs in Henyang City was 3.51 times (95%Cl 55.87–79.72) higher than that in Zhuzhou City, while the infection risk in Yichun City and Fuzhou City was as high as 4.7 (95%Cl 69.93–80.92) and 4.15 (95%Cl 63.81–82.25) times higher compared to Xinyu City (Table 3).

### 3.2. Species/Subtypes and Mixed Infection of Entamoeba spp. in Diarrheic Pigs

In this study, two species (*E. suis*, 17.6%, 129/732; *E. polecki*, 82.4%, 603/732) of *Entamoeba* spp. were observed, and two subtypes (ST1, 85.8%, 573/668; ST3, 35.4%, 357/668) of *E. polecki* were identified. Statistical analysis revealed that *E. polecki* was the predominant species in diarrheic pigs and ST1 was the predominant subtype. The proportion of single infections (58.5%, 428/732, *E. polecki* ST1, *E. polecki* ST3 and *E. suis*) was higher than mixed infections (41.5%, 304/732, *E. polecki* ST1 + *E. polecki* ST3, *E. polecki* ST1 + *E. suis*, *E. polecki* ST3 + *E. suis*, *E. polecki* ST1 + *E. polecki* ST3 + *E. suis*), with ST1 (63.7%, 297/438) being the most frequently identified form of single infection, while *E. polecki* ST1 + *E. polecki* ST3 (78.6%, 239/304) was the most frequently identified form of mixed infection. Two subtypes of *E. polecki* and *E. suis* were observed in Hunan, Jiangxi and Hunan provinces, and all three single infection forms and more than two mixed forms of infections were present in all study regions. Similarly, *E. polecki* ST1, *E. polecki* ST3 and *E. suis* were detected in all age groups of pigs and three forms of single infections and four mixed forms of infections were detected in all different age pigs herds except fattening pigs. *E. polecki* ST1, *E. polecki* ST3 and *E. suis* were also detected in all 11 sampling cities but only one form of single infection (*E. polecki* ST3) was detected in Zhuzhou and Nanping City, while three forms of single infection and more than two mixed infection forms were present in all the remaining cities, including four mixed infection forms in Ji’an, Jiujiang, Nanchang and Yichun City (Table 2 and Table 3).

### 3.3. Phylogenetic Analysis of Entamoeba spp. Isolates in Diarrheic Pigs

Phylogenetic analysis suggested that the sequences of the two subtypes (ST1 and ST3) of *E. polecki* and the sequences of *E. suis* obtained from diarrheic pigs in this study clustered into a branch with other *E. polecki* subtypes (ST1 and ST3) and *E. suis* sequences obtained from other animals (including pigs) or humans, respectively, with higher bootstrap values. In particular, attention should be drawn to the fact that the ST1 sequences (MZ203509-MZ203510) isolated in this study were genetically close to the ST1 sequence (FR686383) isolated from humans, indicating that *E. polecki* ST1 has zoonotic potential, while the sequences of *E. suis* isolated in this study were genetically distant from the sequence of *E. suis* previously isolated from pigs, despite being clustered together (Figure 2).

## 4. Discussion

Although *Entamoeba* spp. is a critical zoonotic parasite, relevant reports of *Entamoeba* spp. infection in pigs based on molecular methods wase very limited, which greatly restricted the understanding of its genetic characteristics and pathogenicity. Therefore, the infection of *Entamoeba* spp. in diarrheic pigs was detected based on molecular means to evaluate the zoonotic potential and further study the genetic structure of *Entamoeba* spp. in pigs.

Based on the SSU rRNA gene of *Entamoeba* spp., the overall infection rate of *Entamoeba* spp. in diarrheic pigs was 58.4% in Hunan, Jiangxi and Fujian provinces. Compared the infections of *Entamoeba* spp. with other studies in healthy pigs, it was higher than in Korea (3.7%) [32], India (50%) [33], Greece (8.3%) [34], Ireland (8% for *E. suis* and 17% for *E. polecki*) [29], Cambodia (31.6%) [35], Germany (52%) [36] and the UK (0%) [37]. It also was higher than that in Vietnam (91.7%) [37], Anhui Province (45.8%) [14] and Fujian Province (55.4%) [25], but lower than that in Tibet (37.9%) [38] and Indonesia (81.1% for *E. suis* and 35.7% for *E. polecki*) [39]. Large differences in the infection of *Entamoeba* spp. within different countries/regions can be found, which may be due to many factors such as different detection methods, different detection kits, geographical factors, personal handling, etc. Furthermore, the sampling time was also an essential factor that cannot be ignored [34,40]. The severe infection of *Entamoeba* spp. was likely attributed to the lack of knowledge of farmers about *Entamoeba* spp. and failure to use drugs for prevention. Although the pathogenicity of most *Entamoeba* spp. was limited, a widespread outbreak of mixed infections with other pathogens would cause serious economic losses and public health problems [9]. Given that the infection differences of *Entamoeba* spp. in the Fujian Province obtained in this study and previously reported were not significant, it may indicate that the effect of diarrhea on *Entamoeba* spp. infection in pigs was relatively limited [25]. No fecal samples from healthy pigs were collected in the same area and at the same time, so there was no way to test healthy pigs and diarrheic pigs for *Entamoeba* spp. infection, making it impossible to directly compare the correlation between diarrhea and the infection, which was the biggest weakness of this study. However, for other hosts, a correlation between diarrhea and infection of *Entamoeba* spp. has been found [41]. This batch of samples was tested simultaneously for other diarrheal pathogens, such as *Blastocystis* sp., *Cryptosporidium* spp., PEDV and PECoV, but unfortunately, the main factors causing diarrhea in pigs were not identified [42,43,44,45]. There were too many factors that could cause diarrhea in pigs, and thus other pathogens are planned to be tested in the future.

In this study, the highest infection rate of *Entamoeba* spp. was observed in sows with 73.4% (406/553, 95%Cl 69.74–77.10), which was 4.42 (95%Cl 3.29–5.96) times higher than the risk of *Entamoeba* spp. infection in suckling piglets. However, in previous reports of healthy pigs for *Entamoeba* spp. infection, the infection rate was highest in weaned piglets and relatively low in sows, which was somewhat different from our results [14,25]. Factors such as breeding management, sample size, sampling time and testing method might be significant reasons for the differences in infection. Moreover, there were some differences in feeding management patterns in different regions, and the density of pigs could have a greater impact on the infection of intestinal protozoa. Whether diarrhea had a great effect on the infection of *Entamoeba* spp. in different age groups of pigs, leading to a relatively high infection rate in sows and fattening pigs, requires more evidence from the future. For older pigs, less attention is usually paid to them because the animals have a better immune system, while the young usually receive a more comfortable living environments and more careful care, which may also be one of the reasons for the greater difference in infection in different age groups. Notably, the infection in healthy pigs was also found to be lowest in suckling piglets, consistent with the results of this study [14,25]. This may be strongly influenced by maternal antibodies.

Two *E. polecki* subtypes (ST1 + ST3) and *E. suis* were observed in pigs from Hunan, Jiangxi and Fujian provinces and all age groups of pigs. The high colonization suggests that pigs may be suitable hosts for the two species of *Entamoeba* spp. The highest infection was observed for *E. polecki* and lower for *E. suis*, and the highest infection was observed for *E. polecki* ST1 among the two subtypes of *E. polecki*, both of which were in full agreement with previous reports. However, the colonization of different genotypes/species of *Entamoeba* spp. at different ages revealed that the results of the present study did not correspond exactly to the previous results [14,25]. The two subtypes of *E. polecki* and *E. suis* in this study were mainly colonized in sows, while previous reports found that these two subtypes were mainly colonized in weaned piglets, with only *E. suis* mainly colonizes sows [25]. Geographical factors, sampling time, sample size, sample type and sanitary conditions were all critical reasons for the differences in infection [46]. According to the data, the investigations of *Entamoeba* spp. infections in pigs were very limited, which poses a great challenge to revealing the actual epidemiological distribution of *Entamoeba* spp. In the present study, the mixed infection of *Entamoeba* spp. was severe, with the main form of mixed infection being *E. polecki* ST1 + *E. polecki* ST3, in agreement with the results of previous studies [25]. The current study found that four mixed forms of infection were observed in all age groups except fattening pigs, whereas, in comparison to previous reports, the four mixed forms of infection were only observed in sows; whether diarrhea was the culprit for this outcome needs further research [25]. However, one of the reasons for the severity of mixed infections must be the poor understanding of the pathogen and the lack of precautions and treatment.

Phylogenetic analysis showed that the sequences of *E. polecki* ST1, *E. polecki* ST3 and *E. suis* obtained in this study all clustered with other sequences of *E. polecki* ST1, *E. polecki* ST3 and *E. suis* sequences, respectively. Notably, the *E. polecki* ST1 sequence (MZ203509 and MZ203510) obtained in this study and other *E. polecki* ST1 sequences (AF149913, MH011333 and MK357717) isolated from pigs and *E. polecki* ST1 sequence (FR686383) isolated from humans, have a close genetic relationship, suggesting that *E. polecki* ST1 has zoonotic potential and that humans may be infected with *E. polecki* ST1 through pigs. This may indicate that geographic isolation has less effect on the genetic variation of *E. polecki* ST1 and the genetic variation of *E. polecki* ST1 in different hosts was equally small, but the validity of the speculation needs to be determined by more detailed sequence analysis. The same result was found in *E. polecki* ST3 sequences. Although human infection with *E. polecki* ST3 has been reported previously, we failed to find the *E. polecki* ST3 sequences and could not further evaluate its zoonotic potential. For *E. suis*, the sequences obtained in this study were found to be clustered into a single branch, whereas the *E. suis* sequences previously isolated from pigs were sister groups to each other, which is similar to the previously reported results and may indicate that *E. suis* isolated in this study may be genetically distant from the previous sequences of *E. suis* [47].

## 5. Conclusions

In this study, the infection of *Entamoeba* spp. in diarrheic pigs in three southern provinces of China was investigated based on the SSU rRNA gene, and a high infection rate was found (58.4%, 732/1254). In addition to the observation of two zoonotic potential subtypes (ST1 and ST3) of *E. polecki* and *E. suis*, severe mixed infections were found. This warns that adequate attention should be paid to *Entamoeba* spp., and appropriate prevention and treatment measures should be developed in a timely manner to reduce economic losses and improve public health. In this study, the species/subtype distribution of *Entamoeba* spp. and the level of infection in pigs with diarrhea were brought to light, the effects of geographic factors and age on infection differences were analyzed and the zoonotic potential of *E. polecki* was assessed, which provides a reference for further studies on the genetic structure of *Entamoeba* spp.

## Figures and Tables

**Figure 1 animals-12-01764-f001:**
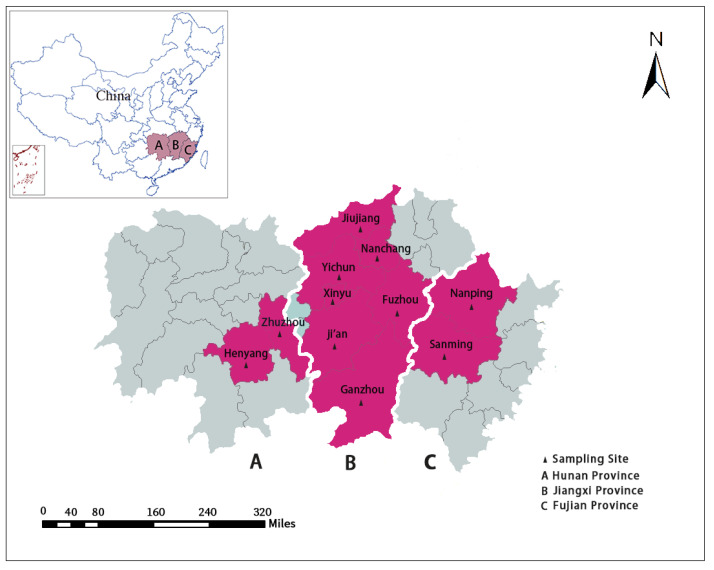
The map of the sample collection in this study, where the red area represents the sampling location.

**Figure 2 animals-12-01764-f002:**
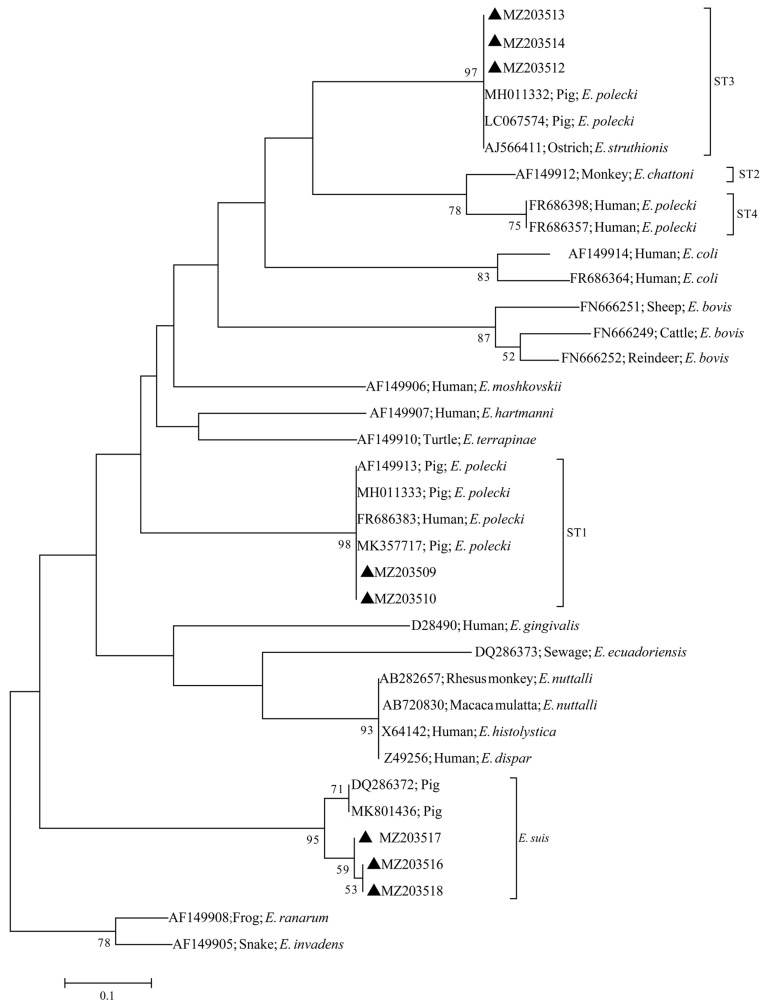
Phylogenetic analysis of *Entamoeba* spp. using maximum likelihood (ML) method based on the SSU rRNA gene. The numbers on the branches represent percent bootstrapping values from 1000 replicates, with values of more than 50% shown in the tree. Marked with solid black triangles are species/subtypes identified in this study.

**Table 1 animals-12-01764-t001:** Factors associated with prevalence and genetic characterizations of *Entamoeba* spp.

Factor	Category	No. Tested	No. Positive (%) [95%CI]	OR (95%Cl)	*p*-Value	Species/Subtype (No.)
Region	Fujian	135	67 (49.63) [41.20–58.06]	Reference	>0.05	ST1 (27), ST3 (17), *E. suis* (2), ST1 + ST3 (19), ST1 + *E. suis* (19)
	Jiangxi	1036	616 (59.46) [56.47–62.45]	1.49 (1.04–2.13)		ST1 (227), ST3 (66), *E. suis* (57), ST1 + ST3 (205), ST1 + *E. suis* (28), ST3 + *E. suis* (10), ST1 + ST3 + *E. suis* (23)
	Hunan	83	49 (59.04) [48.46–69.62]	1.46 (0.84–2.54)		ST1 (25), ST3 (2), *E. suis* (5), ST1 + ST3 (19), ST1 + *E. suis* (2)
Age	Suckling piglets (<21 d)	307	118 (38.44) [33.00–43.88]	Reference	<0.001	ST1 (40), ST3 (14), *E. suis* (20), ST1 + ST3 (36), ST1 + *E. suis* (5), ST3 + *E. suis* (1), ST1 + ST3 + *E. suis* (2)
	Weaned piglets (21–70 d)	299	153 (51.17) [45.50–56.84]	1.68 (1.22–2.32)		ST1 (61), ST3 (27), *E. suis* (12), ST1 + ST3 (39), ST1 + *E. suis* (7), ST3 + *E. suis* (3), ST1 + ST3 + *E. suis* (4)
	Fattening pigs (71–180 d)	95	55 (57.89) [47.97–63.82]	2.20 (1.38–3.52)		ST1 (18), ST3 (5), ST1 + ST3 (31), ST1 + *E. suis* (1)
	Sows (>180 d)	553	406 (73.42) [69.74–77.10]	4.42 (3.29–5.96)		ST1 (160), ST3 (85), *E. suis* (32), ST1 + ST3 (133), ST1 + *E. suis* (19), ST3 + *E. suis* (6), ST1 + ST3 + *E. suis* (17)
Total		1254	732 (58.37) [55.64–64.10]			ST1 (279), ST3 (85), *E. suis* (64), ST1 + ST3 (239), ST1 + *E. suis* (32), ST3 + *E. suis* (10), ST1 + ST3 + *E. suis* (23)

Note: ST1 = *E. polecki* ST1; ST3 = *E. polecki* ST3.

**Table 2 animals-12-01764-t002:** Details of all primers in this study.

Species	Sequence of Primers (5′-3′)	Size (bp)	Annealing Temp (°C)	Reference
*E. histolytica*	E-1: TAAGATGCAGAGCGAAA		56	[7]
	E-2: GTACAAAGGGCGGGACGTA			
	EH-1: AAGCATTGTTTCTAGATCTGAG	439	48	
	EH-2: AAGAGGTCTAACCGAAATTAG			
*E. suis*	764: ATCAAATCAATTAGGCATAACTA		56	[3]
	RD3: ATCCTTCCGCAGGTTCACCTAC			
	764: ATCAAATCAATTAGGCATAACTA	320	52	
	765: AATTAAAACCTTACGGCTTTAAA			
*E. polecki*	Epolec F6: AAATTACCCACTTTTAATTTAGAGAGG	430	55	[8]
	Epolec R6: TTTATCCAAAATCGATCATGAATTTT			
	Epolecki 1: TCGATATTTATATTGATTCAAATG	201	55	[25]
	Epolecki 2: CCTTTCTCCTTTTTTTATATTAG			
	EpST3 F1: GTCTATTCGATCAATTCAATTA	190	45	[4]
	EpST3 R2: TATATTAGTCTTTTTAAAAACTATA			

**Table 3 animals-12-01764-t003:** Distribution of *Entamoeba* spp. in different sampling cities.

Province	City	No. Tested	No. Positive (%) [95%CI] OR (95%Cl)	OR (95%Cl)	*p*-Value	Species/Subtype (No.)
Fujian	Sanming	124	61 (49.19) [40.39–57.99]	Reference	>0.05	ST1 (23), ST3 (17), *E. suis* (1), ST1 + ST3 (18), ST1 + *E. suis* (2)
	Nanping	11	6 (54.55) [25.12–83.97]	1.24 (0.36–4.27)		ST1 (4), *E. suis* (1), ST1 + ST3 (1)
Jiangxi	Xinyu	76	30 (39.47) [28.48–50.46]	Reference	<0.001	ST1 (18), ST3 (4), *E. suis* (4), ST1 + ST3 (2), ST1 + *E. suis* (2)
	Ji’an	185	92 (49.73) [42.52–56.93]	1.52 (0.88–2.61)		ST1 (36), ST3 (9), *E. suis* (5), ST1 + ST3 (25), ST1 + *E. suis* (7), ST3 + *E. suis* (3), ST1 + ST3 + *E. suis* (7)
	Jiujiang	239	119 (49.79) [43.45–56.13]	1.52 (0.90–2.57)		ST1 (35), ST3 (20), *E. suis* (11), ST1 + ST3 (40), ST1 + *E. suis* (8), ST3 + *E. suis* (1), ST1 + ST3 + *E. suis* (4)
	Ganzhou	38	23 (60.53) [44.99–76.07]	2.35 (1.06–5.22)		ST1 (8), ST3 (4), *E. suis* (3), ST1 + ST3 (7), ST3 + *E. suis* (1)
	Nanchang	173	109 (63.01) [55.81–70.20]	2.61 (1.50–4.54)		ST1 (34), ST3 (8), *E. suis* (13), ST1 + ST3 (40), ST1 + *E. suis* (6), ST3 + *E. suis* (3), ST1 + ST3 + *E. suis* (5)
	Fuzhou	89	65 (73.03) [63.81–82.25]	4.15 (2.15–8.01)		ST1 (20), ST3 (7), *E. suis* (3), ST1 + ST3 (33), ST1 + *E. suis* (1), ST3 + *E. suis* (1)
	Yichun	236	178 (75.42) [69.93–80.92]	4.71 (2.72–8.13)		ST1 (76), ST3 (14), *E. suis* (18), ST1 + ST3 (58), ST1 + *E. suis* (4), ST3 + *E. suis* (1), ST1 + ST3 + *E. suis* (7)
Hunan	Zhuzhou	24	9 (37.5) [18.13–56.87]	Reference	<0.05	ST1 (3), *E. suis* (4), ST1 + ST3 (2)
	Hengyang	59	40 (67.80) [55.87–79.72]	3.51 (1.30–9.45)		ST1 (22), ST3 (2), *E. suis* (1), ST1 + ST3 (13), ST1 + *E. suis* (2)
Total		1254	732 (58.37) [55.64–64.10]			ST1 (279), ST3 (85), *E. suis* (64), ST1 + ST3 (239), ST1 + *E. suis* (32), ST3 + *E. suis* (10), ST1 + ST3 + *E. suis* (23)

Note: ST1 = *E. polecki* ST1; ST3 = *E. polecki* ST3.

## Data Availability

Not applicable.

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
