# Peer review of "Molecular Characterization of Entamoeba spp. in Pigs with Diarrhea in Southern China"

_animals, 2022, doi:10.3390/ani12141764_

Round 1
Reviewer 1 Report
This manuscript reports on the molecular epidemiology of Entamoeba spp. in fecal samples of diarrheic pigs from three provinces in southern China. Entamoeba is a zoonotic intestinal protozoa that can parasitize most invertebrates including pigs and humans.
General comments: The manuscript is well written. There are minor typographical errors that should be addressed. The methods should be improved. Specific comments are as follows:
Abstract
Line 34: Italicize species names
Introduction
Line 34-35: Check that the explanation of free-living and parasitic species of Entamoeba are not reversed.
Line 53-54: The sentence should be revised for clarity.
Line 90: What is "spp. species"?
Materials and Methods
How was the sample size determine? Proper description of the sample size determination will greatly improve the quality of this manuscript and support the arguments in the discussion section.
Line 132 - 133: Revise the sentence for clarity
Line 147: Check sentence
Results
Line 236: Check species name formatting
Discussion
Line 281: Revise sentence for clarity
Line 304 - 306: The sentence "Geographical factors, sampling time, sample size, sample type, and sanitary conditions were all critical reasons for the differences of infection" does not have proper backing from the methods or the results section. Explain the effects of sample size? What were the sampling times? What were the other sample types either than fecal samples? Explain the variability in the sanitary conditions. A proper description of the sample size determination and the sampling procedure should be provided in the methods section.
Author Response
On behalf of all co-authors, I would like to thank you and the two reviewers very much for positive comments and constructive suggestions on our manuscript (MS) ID animals-1790733. The reviewers considered our MS to be of general interest to the readership of Animals, and recommended the acceptance of our MS for publication after revisions.
Therefore, we have revised the MS strictly according to the reviewers’ comments and suggestions. We used the “tracked changes” mode in the WORD to show the revised/changed text in the revised MS. Two MS files are uploaded: the “clean version” as “manuscript”, and the one showing “tracked changes” as “supplementary material”. In the following, we detail our point-by-point responses to the reviewer’s comments and suggestions. We made all our responses in red colour for clarity.

Reviewer 2 Report
Entamoeba spp. is a common zoonotic intestinal protozoan that can parasitize most vertebrates, including humans and pigs, causing severe intestinal diseases and posing a serious threat to public health. However, the available data on Entamoeba spp. infection in pigs is relatively limited in China. This manuscript found that Entamoeba polecki and Entamoeba suis were detected. The ST1 and ST3 subtypes of Entamoeba polecki were detected, and a relatively serious mixed infection was found. These findings provide baseline data for preventing and controlling Entamoeba spp. infection in southwestern China.
1. Please review the grammar English of this manuscript.
2. Please find these papers: Zhang Q, et al. Acta Parasitol. 2019. PMID: 30671772; Al-Habsi K, et al. Vet Parasitol. 2017. PMID: 28215866; Candela E, et al. Parasit Vectors. 2021. PMID: 34598722;
Author Response

(The authors gave the same response as above.)

Round 2
Reviewer 2 Report
Thank you very much! Accepted it.